# Learning Global Transparent Models Consistent with Local Contrastive Explanations

**Tejaswini Pedapati**
IBM Research
tejaswinip@us.ibm.com

**Avinash Balakrishnan**
IBM Research
avinash.bala@us.ibm.com

**Karthikeyan Shanmugan**
IBM Research
karthikeyan.shanmugam2@ibm.com

**Amit Dhurandhar**
IBM Research
adhuran@us.ibm.com

## Abstract

There is a rich and growing literature on producing local contrastive/counterfactual explanations for black-box models (e.g. neural networks). In these methods, for an input, an explanation is in the form of a contrast point differing in very few features from the original input and lying in a different class. Other works try to build globally interpretable models like decision trees and rule lists based on the data using actual labels or based on the black-box models predictions. Although these interpretable global models can be useful, they may not be consistent with local explanations from a specific black-box of choice. In this work, we explore the question: Can we produce a transparent global model that is simultaneously accurate and consistent with the local (contrastive) explanations of the black-box model? We introduce a natural local consistency metric that quantifies if the local explanations and predictions of the black-box model are also consistent with the proxy global transparent model. Based on a key insight we propose a novel method where we create custom boolean features from sparse local contrastive explanations of the black-box model and then train a globally transparent model on just these, and showcase empirically that such models have higher local consistency compared with other known strategies, while still being close in performance to models that are trained with access to the original data.

## 1 Introduction

With the ever increasing adoption of black-box artificial intelligence technologies in various facets of society [12], a number of explainability algorithms have been developed to understand their decisions. Two prominent types are: a) Local Explanations for a data point of interest [16, 20, 22, 25, 9] and b) Constructing interpretable global models directly like decision trees, rule lists and boolean rules [23, 10, 7]. One of the arguments for b) is that it is possible to construct interpretable global models in such a way that for a single data point it can give a succinct local explanation in the form of a sparse conjunction [23] while also highlighting non-trivial global behavior of the output. Methods in a) naturally enjoy the parsimonious explanations on a single data point through either feature importance scores or contrastive points that differ in very few features. Therefore, one generally has the option of adopting an accurate black-box model (viz. Neural Network, XGBoost) with local explanations obtained with additional computational effort or adopt a simpler but interpretable model that can describe non local behavior as well as provide very fast local explanations. For tabular data there is a growing set of techniques to build models of the later kind directly [23] that can compete in

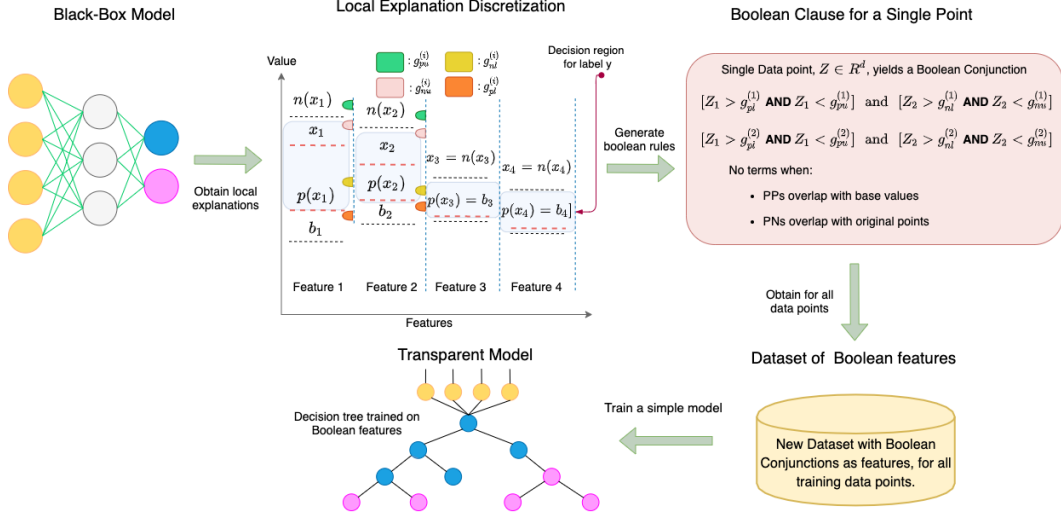

Figure 1: Above we see the main steps behind our approach, where the key idea is to make the generated boolean clause match with predictions on the *triplet* (original point $\mathbf{x}$, its PP $\mathbf{p}(\mathbf{x})$, its PN $\mathbf{n}(\mathbf{x})$) of the black-box (i.e. match input prediction and local explanation). To do this we first obtain local explanations, then perform a custom discretization and binning obtaining upper and lower bounds for PPs ($g_{pu}^{(i)}$, $g_{pl}^{(i)}$) and PNs ($g_{nu}^{(i)}$, $g_{nl}^{(i)}$) producing the final boolean clauses. $g_{pl}^{(i)}$ and $g_{nu}^{(i)}$ are just rounding of the PP and the PN values for feature $i$ to the nearest grid point. While $g_{pu}^{(i)}$ and $g_{nl}^{(i)}$ ensure that the clause would be local to $x$. A new dataset with clauses created as shown above can serve as input to a simple model training algorithm.

terms of accuracy with complex black-box models. However, black-box models have demonstrated performance in terms of higher accuracy and in many cases are still preferred [21, 1].

In domains such as finance and healthcare, one would like to offer explanations (local and global) that are loyal to the model deployed. Proxy models that provide global interpretability [11, 10, 5] may not capture the original model's local behavior. This might be important in many practical applications [19, 2], where we want the global interpretable proxy model to replicate the local behavior of the black-box model as much as possible. Given this need, we in this paper address the following two important questions: *a) How do we measure if a simple model replicates the local behavior of a given black-box? b) Given access to a black-box model and a local explainbility method, how do we better train simple models belonging to a specific class that are accurate and good at mimicing the local behavior of the black-box model?* Moreover, we want to answer b), where we primarily have access to local sparse explanations.

We consider local explanation methods [29, 8] that produce contrast point(s) for every input point. Contrast points for a given data point are two different perturbations of the point such that the number of features altered are minimal. The first type of perturbation retains minimal amount of information from the original point so that the new point is still classified by the model in the same class and is termed as a pertinent positive (PP). The second is a perturbation of the input point which would make it classify into a different class and is termed as a pertinent negative (PN). For example in a loan approval application, say a person's loan was rejected whose debt was 30,000, age was 27 and salary was 50,000. If reducing their debt to 20,000 still resulted in a rejection, where other factors had to remain the same, then the profile of debt 20,000, age 27 and salary 50,000 would be a PP. If on the other hand, their salary increasing to 70,000 with other things being the same, led to the loan being approved, then the resultant point would be a PN.

With this, to address the first question above we propose a natural consistency metric where we check if a candidate transparent model agrees in the classification of the original point along with the contrast points produced by a local explanation method for some black-box model. To address the second question, one of our key insights is that each sparse local explanation in the form of contrast points can be transformed into a *logical conjunction of conditions on few features* with a

custom discretization scheme. Ideally, an approach of building a global transparent model from local explanations of a black-box model would retain the structure (sparse interaction of various features) of the local explanations.

We thus propose a new algorithm that uses local explanations from a contrastive explanations method to generate boolean clauses which are conjunctions. These boolean conjunctions can be used as features, forming a new dataset, to train another simple model such as logistic regression or a small decision tree. An illustration of the whole process that we just described is given in Figure 1. The algorithm binarizes local contrastive explanations depending on the difference in feature values between the contrast point and the original. The binarization of features is directed by local explanations based on ranges that are deemed locally important by the black-box model. One of the most interesting aspects of this idea is that sparse interactions between original features required for explaining, are directly captured by these boolean clauses that can be used to train transparent models that generalize and thus effectively capture also global information.

## 2    Related Work

Most of the work on explainability in artificial intelligence can be said to fall under four major categories: Local posthoc methods, global posthoc methods, directly interpretable methods and visualization based methods.

**Local Posthoc Methods:** Methods under this category look to generate explanations at a per instance level for a given black-box classifier that is uninterpretable. Methods in this category are either proxy model based [22, 24] or look into the internals of the model [3, 8, 29, 20]. Some of these methods also work with only black-box access [22, 9]. There are also a number of methods in this category specifically designed for images [26, 3, 25].

**Global Posthoc Methods:** These methods try to build an interpretable model on the whole dataset using information from the black-box model with the intention of approaching the black-box models performance. Methods in this category either use predictions (soft or hard) of the black-box model to train simpler interpretable models [11, 4, 5] or extract weights based on the prediction confidences reweighting the dataset [10].

**Directly Interpretable Methods:** Methods in this category include some of the traditional models such as decision trees or logistic regression. There has been a lot of effort recently to efficiently and accurately learn rule lists [23] or two-level boolean rules [28] or decision sets [27]. There has also been work inspired by other fields such as psychometrics [14] and healthcare [6].

**Visualization based Methods:** These try to visualize the inner neurons or set of neurons in a layer of a neural network [13]. The idea is that by exposing such representations one may be able to gauge if the neural network is in fact capturing semantically meaningful high level features.

The most relevant categories to our current endeavor are possibly the local and global posthoc methods. The global posthoc methods although try to capture the global behavior of the black-box models, the coupling is weak as it is mainly through trying to match the output behavior, and they do not leverage or are necessarily consistent with the local explanations one might obtain.

## 3    Preliminaries

To learn our boolean features we first need a local explainability technique that can extract contrastive explanations for us from arbitrary black-box models. The method we use is the model agnostic contrastive explanations method [9], which generates PPs (pertinent positives) and PNs (pertinent negatives) with just black-box access. Both PPs and PNs are contrast points that preserve or change labels of the blackbox model with respect to the original point.

**Base Value Vector:** To find PPs/PNs the method in [9] requires specifying values for each feature that are least interesting, which they term as base values. A user can prespecify semantically meaningful base values or a default value of say median could be set as a base value. Classifiers essentially pick out the correlation between variation from this value in a given co-ordinate and the target class $y$. Therefore, we define a vector of base values $\mathbf{b} \in \mathbb{R}^d$. Variation away from the base value is used to correlate with the target class. $b_j$ represents the base value of the $j$-th feature.

**Upper and Lower Bounds:** Let $L_j$ and $U_j$ be lower and upper bounds for the $j$-th feature $x_j$.

Consider a pre-trained classifier $\zeta$ and let $C(\mathbf{x}, y)$ denote the classifiers confidence score (a probability) of class $y$ given $\mathbf{x}$.

**Pertinent Positive**: Let $\mathbf{p}(\mathbf{x}) \in \mathbb{R}^d$ denote the pertinent positive vector associated with a training sample $(\mathbf{x}, y) \in \mathcal{D}$.

$$\mathbf{p}(\mathbf{x}) = \underset{\delta}{\arg\min} \max\{\max_{\tilde{y} \neq y} C(\delta, \tilde{y})\} - C(\delta, y), -\kappa\}$$

$$\text{s.t.} \quad L_j \leq \delta_j \leq U_j \text{ and } \|\delta - \mathbf{b}\|_0 \leq k \text{ and } \delta_j \leq x_j, \ j : x_j \geq b_j \text{ and } \delta_j \geq x_j, \ j : x_j \leq b_j \quad (1)$$

In other words, Pertinent positive is a sparse vector that the classifier $\zeta$ classifies in the same class as the original input with high confidence. Being sparse, it is expected to have lesser variation away from the base values than $\mathbf{x}$.

**Pertinent Negative**: Let $\mathbf{n}(\mathbf{x}) \in \mathbb{R}^d$ denote the pertinent positive vector associated with a training sample $(\mathbf{x}, y) \in \mathcal{D}$.

$$\mathbf{n}(\mathbf{x}) = \underset{\mathbf{x}+\delta}{\arg\min} \max\{C(\mathbf{x} + \delta, y) - \max_{\tilde{y} \neq y}\{C(\mathbf{x} + \delta, \tilde{y})\}, -\kappa\}$$

$$\text{s.t.} \quad L_j \leq \delta_j \leq U_j \text{ and } \|\delta\|_0 \leq k \text{ and } \delta_j \geq 0, \ j : x_j \geq b_j \text{ and } \delta_j \leq 0, \ j : x_j \leq b_j \quad (2)$$

In other words, Pertinent negative is a vector where there are few coordinates which are different from $\mathbf{x}$. It forces the classifier $\zeta$ to classify it in some other class with high confidence, and in coordinates where it differs from $\mathbf{x}$, those coordinates are farther from the base values than those of $\mathbf{x}$.

# 4  Methodology

In this section, we first define a local consistency metric that quantifies the extent to which a transparent model captures the local behavior of a black-box model. This is followed by a description of our proposed method.

## 4.1  Local Consistency Metric

Consider a dataset consisting of samples $(\mathbf{x}, y) \in \mathcal{D}$. $\mathbf{x} \in \mathbb{R}^d$ and $y \in \mathcal{Y}$. $\mathcal{Y}$ is a finite set of class labels. Let $\zeta_B(.)$ and $\zeta_T(.)$ denote the black-box and transparent classifiers respectively that map inputs to labels ($\mathbb{R}^d \rightarrow \mathbb{Z}$). Let $\mathbf{p}(\mathbf{x}) \in \mathbb{R}^d$ and $\mathbf{n}(\mathbf{x}) \in \mathbb{R}^d$ denote the PP and PN for sample $(\mathbf{x}, y)$ based on $\zeta_B(.)$. Note that this implies that $\zeta_B(\mathbf{x}) = \zeta_B(\mathbf{p}(\mathbf{x}))$ and $\zeta_B(\mathbf{x}) \neq \zeta_B(\mathbf{n}(\mathbf{x}))$. Let $\lambda_{TB}(.)$ then denote the following loss function:

$$\lambda_{TB}(\mathbf{x}) = \begin{cases} 0, & \text{if } \zeta_B(\mathbf{x}) = \zeta_T(\mathbf{x}) \text{ and } \zeta_B(\mathbf{x}) = \zeta_T(\mathbf{p}(\mathbf{x})) \text{ and } \zeta_B(\mathbf{x}) \neq \zeta_T(\mathbf{n}(\mathbf{x})) \\[1em] 1, & \text{otherwise} \end{cases} \quad (3)$$

The above loss for a sample $\mathbf{x}$ is thus zero only if i) the transparent model has the same prediction as the black-box model, ii) the PP prediction for $\zeta_T(.)$ matches the prediction of $\zeta_B(.)$ on the original sample and iii) the PN prediction for $\zeta_T(.)$ is different than the prediction of $\zeta_B(.)$ on the original sample. Cases ii) and iii) together ensure that the transparent model is consistent with the local explanation of the black-box model. Given this per-sample loss and if $|.|$ denotes cardinality, we now define the local consistency metric $\mathcal{C}_{TB}$ for a transparent model $\zeta_T$ w.r.t $\zeta_B$ as:

$$\mathcal{C}_{TB} = 1 - \frac{1}{|\mathcal{D}|} \sum_{(\mathbf{x}, y) \in \mathcal{D}} \lambda_{TB}(\mathbf{x}) \quad (4)$$

The above metric is a natural way of measuring local consistency as it takes into account not just the prediction of the black-box model, but also its behavior around each sample.

## 4.2  Generating Sparse Boolean Clauses from Pertinent Positives and Pertinent Negatives

The following is the key observation of our work that lets us mine interpretable Boolean features.

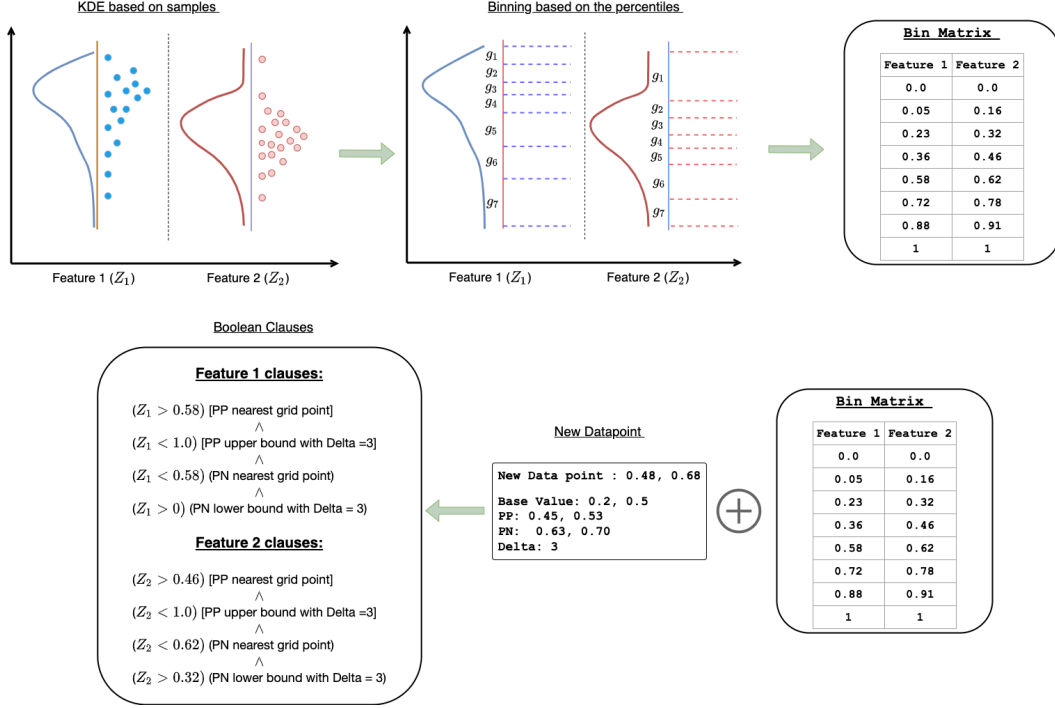

Figure 2: An Illustration of how binning works (top row). A boolean rule generated for a datapoint is also demonstrated (bottom row). The rules are thus formed by looking at the PP, PN (local explanation) and the bin matrix and $\Delta$.

**Key Idea:** For a given training point $\mathbf{x}$, we observe that $k$-sparse PPs and PNs give rise to a $2k$-sparse Boolean AND clause, as we describe next. Note that this implies that the complexity of our boolean features is tightly bounded by the complexity of the local explanations, which are likely to be sparse, and hence, much more concise than the size of the input dimension.

Pertinent positives says simply that a specific feature has to have at least the variation of the pertinent positive feature value for it to be classified into that class. Similarly, pertinent negatives say that a specific feature can have a variation more than $x_j$ *but not* beyond $\delta_j$. We have illustrated this key idea in Figure 1.

Therefore, one can come up with the following Boolean Clause written as product of indicators:

$$\left[ \prod_{j:p[x_j]>b_j} \mathbf{1}(z_j > p[x_j]) \right] \left[ \prod_{j:p[x_j]<b_j} \mathbf{1}(z_j < p[x_j]) \right] \left[ \prod_{j:n[x_j]>b_j} \mathbf{1}(z_j < n[x_j]) \right] \left[ \prod_{j:n[x_j]<b_j} \mathbf{1}(z_j > n[x_j]) \right]$$

(5)

**Regularizing by bounds and binning using grid points:** In practice, these explanations are very local and hence adding further bounds will help in generalization. This is because if for some training example $x$, $\mathbf{1}(z_j > p[x_j])$ is the condition. Then points from some other class can also satisfy this inequality which is an infinite interval on the real line. Since these clauses are derived from local perturbations (because of $\ell_2$ penalty in the optimization), they may not be valid very far from $x_j$. Also we need to reduce the number of distinct clauses. We address both of these issues by binning each feature into multiple grid points, where the grid values closest to each feature value serve as upper and lower bounds that are used by the resultant boolean formula.

In particular, we only use clauses involving grid points $g_{sj}$ where every coordinate $j$ has at most $N+1$ grid points. We will call the matrix of $\{g_{sj}\}$ as the Grid Matrix $G$. We round off all the clauses to the nearest grid points suitably and also add regularizing upper bounds using grid points that are $\Delta$ far away from the grid point involved in the clause. In Figure1, the grid values for features 1 and 2

are shown. In this case $N$ is 3. The closest grid point less than $p(x_1)$ is $g_{pl}$. We describe this in detail below for 2 out of the 4 cases that arise in Algorithm 1.

**Clause added for PP** (refer to Line 8 in Algorithm 1) For some feature $x$, suppose the pertinent positive $p$ is such that $b < p < x$ where $b$ is the base value of the feature. We find two closest grid points as follows: a) $g_i < p$ and closest to $p$ and b) $g_j > x$ that is closest to $x$. Here, $i$ and $j$ are their indices when you sort the grid points from the lowest to the highest. Then, instead of a clause $\mathbf{1}(x > p)$, i.e. that captures the condition that *the feature has to be at least $p$*, we will have conjunction of two clauses: $\mathbf{1}(x > g_i) \times \mathbf{1}(x < g_{j+\Delta})$. This new conjunction says *the feature has to be at least the grid value below and closest to $p$ (due to binning) and it should not be too far away from the original point (due to explicit regularization)*. Here, $\Delta$ is a skip parameter, that we optimize over during cross validation. In Figure1, the PP clause for feature1 would yield $x > g_{pl}^{(1)} \times x < g_{pu}^{(1)}$ When $p < b$, the inequalities flip and an analogous clause is given starting from Line 11.

**Clause added for PN** (refer to Line 16 in Algorithm 1) For a pertinent negative $n$ for a feature satisfying $n > x > b$, we find two grid points $g_i : n > g_i > x$ and $g_i$ is the closest to $n$ and $g_j$ such that $g_j < x$ and $g_j$ is the closest to $x$ and $j < i$. Now, instead of the clause $\mathbf{1}(x < n)$ i.e. that captures the condition that *the feature has to be at most $n$*, we substitute the conjunction $\mathbf{1}(x < g_i) \times \mathbf{1}(x > g_{j-\Delta})$. This new conjunction says *the feature has to be at most the grid value below and closest to $n$ (due to binning) and it should not be too far away from the original point (due to explicit regularization)* When the point $x < b$, the inequalities flips and the clause is given starting from Line 19.

The boolean clause generation algorithm incorporating all these ideas is given in Algorithm 1 that covers all relevant cases of relative ordering between base values, pertinent positives and negatives and the feature values. Some visual intuition about the discretization is given in Figure 1.

---

**Algorithm 1** Global Boolean Feature Learning (GBFL)

---
1: **Input:** Training set:$D$ , Binning Matrix: $G$, Base Values vector: $\mathbf{b}$, Pertinent Positives: $\mathbf{p}(D)$, Pertinent Negatives: $\mathbf{n}(D)$, Skip Parameter: $\Delta$.
2: **Output:** A set of Boolean clauses $\mathcal{F}$.
3: **for** Training sample $(\mathbf{x}, y) \in \mathcal{D}$ **do**
4:     $Q = 1$
5:     **for** $j$ in $[1 : d]$ **do**
6:         **if** $p[x_j] > b_j$ **then**
7:             `** Generating clauses for the PP. The clause depends on whether the PP for a feature is above the base value or not **`
8:             $g* \leftarrow \underset{g_{sj}:b_j<g_{sj}<p(x_j)}{\mathrm{argmin}} |g_{sj} - p(x_j)|, \ s* \leftarrow \underset{s:g_{sj}>x_j}{\mathrm{argmin}} |g_{sj} - x_j|$
9:             $Q \leftarrow Q * \mathbf{1}(z_j \geq g*) * \mathbf{1}(z_j < g_{s*+\Delta})$
10:         **else**
11:             $g* \leftarrow \underset{g_{sj}:b_j>g_{sj}>p(x_j)}{\mathrm{argmin}} |g_{sj} - p(x_j)|, \ s* \leftarrow \underset{s:g_{sj}<x_j}{\mathrm{argmin}} |g_{sj} - x_j|$
12:             $Q \leftarrow Q * \mathbf{1}(z_j < g*) * \mathbf{1}(z_j \geq g_{s*-\Delta})$
13:         **end if**
14:         **if** $x_j > b_j$ **then**
15:             `** Generating clauses for the PN. The clause depends on whether the original feature is above the base value or not **`
16:             $g* \leftarrow \underset{g_{sj}:n(x_j)>g_{sj}>x_j}{\mathrm{argmin}} |g_{sj} - n(x_j)|, \ s* \leftarrow \underset{s:g_{sj}<x_j}{\mathrm{argmin}} |g_{sj} - x_j|$
17:             $Q \leftarrow Q * \mathbf{1}(z_j < g*) * \mathbf{1}(z_j \geq g_{s*-\Delta})$
18:         **else**
19:             $g* \leftarrow \underset{g_{sj}:n(x_j)<g_{sj}<x_j}{\mathrm{argmin}} |g_{sj} - n(x_j)|, \ s* \leftarrow \underset{s:g_{sj}\geq x_j}{\mathrm{argmin}} |g_{sj} - x_j|.$
20:             $Q \leftarrow Q * \mathbf{1}(z_j > g*) * \mathbf{1}(z_j < g_{s*+\Delta})$
21:         **end if**
22:     **end for**
23:     $\mathcal{F} \leftarrow \mathcal{F} \cup Q$
24: **end for**

---

**KDE Binning:** The binning technique used to determine the grid points is an important part of the algorithm. We actually would like to place the grid points such that it creates equally spaced intervals

such that every interval has equal probability. Consider the $j$-th feature $x_j$. Suppose, $L_j$ and $U_j$ are lower and upper bounds for this feature. We estimate the marginal density of this feature using a KDE estimate by using an appropriate kernel with a bandwidth on the points taken from this feature and obtain a cumulative density function $P_j(x)$. Suppose $N + 1$ is the number of grid points we desire. Now, using root finding techniques, we actually find $\frac{k}{N}$-th quantile for $k \in \{1, 2, 3..N - 1\}$. This grid generation procedure is given in Algorithm 2. A toy example is provided in Figure 2.

---

**Algorithm 2** KDE based Grid Point Generation (GPG)

---
.

1: **Input:**  Number of grid points: $N$, Dataset: $D$, Bounds on Features:$\{L_j, U_j\}_j$,Bandwidth for KDE estimator: $B$.
2: **Output:** Grid matrix $G \in \mathbb{R}^{N+1 \times d}$
3: **for** $j$ in $[1 : d]$ **do**
4:     Collect the set of points $\mathcal{P}_j$ from feature $j$ from dataset $D$. Obtain the KDE estimate $p_j(x) = \frac{1}{B|\mathcal{P}_j|} \sum_{p \in \mathcal{P}_j} K(\frac{x-p}{B})$.
5:     Obtain the CDF function for this $P_j(x)$. Set lower and upper bounds to the extreme grid points: $G_{j0} \leftarrow L_j, G_{jN} \leftarrow U_j$.
6:     **for** $n$ in $[1 : N - 1]$ **do**
7:         Use root finding to set $G_{jn} \leftarrow \{x : P_j(x) = \frac{n}{N}, \ L_j \leq x \leq U_j\}$
8:     **end for**
9: **end for**

---

---

**Algorithm 3** Model Generation using Local Explanations

---
.

1: Grid Matrix $G \leftarrow \mathsf{GPG}(N, D, \{L_j, U_j\}_j, B)$
2: Use dataset $D$, a pre-trained black-box model $\zeta$, find pertinent positives $\mathbf{p}(D)$ and negatives $\mathbf{n}(D)$ for every training sample using (1) and (2). Obtain the set of boolean clauses: $\mathcal{F} \leftarrow \mathsf{GBFL}(D, G, \mathbf{b}, \mathbf{p}(D), \mathbf{n}(D))$
3: Form a binary dataset $D_\mathcal{F} \in \{0, 1\}^{|D| \times |\mathcal{F}|}$ whose rows are obtained by evaluating a training point $\mathbf{x} \in D$ through all clauses $C \in \mathcal{F}$.
4: Fit a simple transparent learner to this new dataset: $\mathsf{TL}(D_\mathcal{F}) \rightarrow \zeta_T$.

---

**Learning Algorithm:** We assume that a transparent leaner ($\mathsf{TL}$) is given to us like a Decision Tree learner or a Logistic Regression based learner. Then algorithm 3 learns a transparent model based on the boolean rules/features extracted using GBFL.

## 5 Experiments

We now empirically validate our method. We first describe the setup, followed by a discussion of the experimental results. We provide quantitative results as well as present the most important boolean features picked by the transparent models based on our construction for one of the datasets. A more complete list of important boolean features is provided in the supplement.

**Setup:** We experimented on six publicly available datasets from Kaggle and UCI repository namely; Sky Survey, Credit Card, Magic, Diabetes, Waveform and WDBC. The dataset characteristics are given in the supplement. Deep neural networks (DNNs) with (different) architectures that provided best results were chosen for each of the datasets as the black-box model (details in supplement). Decision trees (DTs) with height $\leq 5$ were the transparent learner based on the CART algorithm. Gridding and $\Delta$ information for GBFL is also provided in the supplement. Statistically significant results that measure performance based on paired t-test are reported, which are computed over 5 randomizations with 75/25% train/test split. 10-fold cross-validation was used to find all parameters including tree heights ($\leq 5$) for DT. *All trees for the competitors ended up being of height strictly $< 5$, which means that we are not restricting the expressive power of the other approaches to build locally consistent models.*

We compared our method against three other approaches. i) *Standard:* Train DT on original dataset, ii) *Distillation/Model Compression [5]:* Train DT on black-box models predictions using original input features and iii) *Augmentation [15]:* Train DT on PPs and PNs obtained for each sample of the original data based on the black-box model augmented to the original data (i.e. data + PPs + PNs). This is similar to [15], which showed the power of such augmentation, where they, although, had human provided counterfactual explanations.

Table 1: Below we see the performance of the different methods on consistency and accuracy metrics. $\mathcal{C}_T^{PN}$ and $\mathcal{C}_T^{PP}$ imply consistency w.r.t. just PNs and PPs respectively. NA implies the explanation method was not able to find PNs for those datasets. Best results based on paired-t test for the transparent model (DT) are in bold. All numbers are percentages.

| Metric | Method | Sky Survey | Credit Card | Magic | Diabetes | Waveform | WDBC |
|---|---|---|---|---|---|---|---|
| $\mathcal{C}_{TB}$ | Standard | 31.08 | 3.51 | 54.04 | 61.97 | 12.52 | 18.79 |
| | GBFL | **48.84** | **7.03** | 56.76 | **68.75** | **19.55** | **50.12** |
| | Distillation | 33.59 | 4.52 | 56.04 | 64.58 | 5.48 | 22.47 |
| | Augmentation | 3.86 | 2.01 | **57.24** | 58.85 | 7.03 | 5.92 |
| $\mathcal{C}_{TB}^{PN}$ | Standard | 44.01 | 49.74 | NA | NA | 59.69 | 29.01 |
| | GBFL | **58.88** | **74.87** | NA | NA | **65.18** | **61.01** |
| | Distillation | 38.22 | 23.11 | NA | NA | 29.15 | 33.03 |
| | Augmentation | 12.16 | 51.75 | NA | NA | 62.77 | 19.02 |
| $\mathcal{C}_{TB}^{PP}$ | Standard | 75.67 | 61.80 | 59.36 | 68.23 | 82.33 | 88.78 |
| | GBFL | **79.44** | 56.23 | **61.68** | **68.75** | **84.66** | 89.67 |
| | Distillation | 78.57 | **66.43** | 59.32 | 68.22 | 72.72 | 79.99 |
| | Augmentation | 75.17 | 57.28 | **61.96** | 66.67 | 80.39 | **91.99** |
| Test Accuracy | Standard | **99.2** | **60.7** | **79.24** | 75.10 | **75.92** | 93 |
| | GBFL | 97.68 | 59.9 | 77.30 | 75 | **75.48** | 93 |
| | Distillation | 92.88 | 48.6 | **79.96** | **77.08** | 61.2 | 92.01 |
| | Augmentation | 96.21 | 58.4 | 77.8 | 71.87 | 74.72 | **95.07** |
| | Black-box | 99.68 | 69.4 | 88.98 | 83 | 85.8 | 96.1 |

**Quantitative Evaluation:** In Table 1, we see that GBFL has higher local consistency than the other competitors in most cases. For Magic and Diabetes where the explanation method we used was not able to generate PNs we calculate $\mathcal{C}_{TB}$ by ignoring the PN condition in equation 3. Interestingly, in these two cases our gain in the consistency over other methods is either absent or relatively low. This implies that our method is quite consistent w.r.t. PNs compared with other approaches. In fact, this is verified by looking at $\mathcal{C}_{TB}^{PN}$ for datasets that we do have PNs (Sky Survey, Credit Card, Waveform and WDBC) where our consistency w.r.t. PNs is much better than the other methods. Consistency w.r.t. PPs ($\mathcal{C}_{TB}^{PP}$) is superior to other methods in most cases, but to a lesser degree.

In addition to being more locally consistent we note that our accuracy is still competitive with the best performing methods in all cases. This is especially promising given that our features are built *only* based on sparse local explanations without access to the original data, while all other methods have access to it.

**Qualitative Evaluation:** We now show boolean features constructed by our method for the Sky Survey dataset (more examples in supplement) using the contrastive explanations for the respective black-box models that the base learner deems as most important. We see that although the boolean features are composed of multiple input/original features, the resultant model is still transparent and reasonably easy to parse. Definitely more so than the original black-box model.

*Sky Survey Dataset:* In Listing 1, we see the top 4 boolean features based on the decision tree which is the base transparent learner. We observe that rules with nine features were created based on the contrastive explanations, which are a union of the PP and PN features using algorithm 1. The different boolean features thus have conditions i.e. upper and lower bounds on the same set of original features.

We can see that some of the original features such as *redshift*, *mjd* and *dec* have the same condition across multiple boolean features indicating that them along with their ranges are likely to be most important. On the other hand, *ra* has different ranges in most cases. The other features have in between redundancy relative to ranges.

Listing 1: We present below the top 4 GBFL features used by DT on the Sky Survey dataset.

```
GBFL rank 1 feature
1.51>= dirf1 >=0.0 & 0.66>= dirf2 >=0.0 & 0.26>= dirf3 >=0.0 &
629.05>= fiberid >=419.36 & 0.80>= redshift >=0.0 & 233.35>= ra >=137.26 &
57481>= mjd >=42354.42 & 14.42>= dec >=0.0 & 1770.52>= plate >=0.0

GBFL rank 2 feature
```

```
2.53 >= d i r f 1 >=0.50 & 0.49 >= d i r f 2 >=0.0 & 0.35 >= d i r f 3 >=0.0 &
2213.15 >= p l a t e >=442.63 & 14.42 >= dec >=0.0 & 0.80 r e d s h i f t >=0.0 &
262.10 >= f i b e r i d >=52.42 & 164.72 >= ra >=109.81 & 57481 >= mjd >=42354.42

GBFL rank 3 f e a t u r e
1.51 >= d i r f 1 >=0.0 & 0.49 >= d i r f 2 >=0.0 & 0.35 >= d i r f 3 >=0.0 &
260.81 >= ra >=233.35 & 0.80 >= r e d s h i f t >=0.0 & 524.21 >= f i b e r i d >=157.26 &
14.42 >= dec >=0.0 & 1770.52 >= p l a t e >=0.0 & 57481 >= mjd >=42354.42

GBFL rank 4 f e a t u r e
2.53 >= d i r f 1 >=0.50 & 0.49 >= d i r f 2 >=0.0 & 0.44 >= d i r f 3 >=0.089 &
576.63 >= f i b e r i d >=366.94 & 260.81 >= ra >=233.35 & 14.42 >= dec >=0.0 &
0.80 >= r e d s h i f t >=0.0 & 1770.52 >= p l a t e >=0.0 & 57481.0 >= mjd >=42354.42
```

# 6   Discussion

As systems get more complicated replicating their behavior using simple interpretable models might become increasingly challenging. Transparent models could be the answer here where there is more leeway to build more complicated models that can be traced for the decisions they make and hence are auditable. Auditability is extremely important in domains such as finance, where decisions need to be traceable and proxy models to explain black-boxes need to have stronger guarantees that they truly are replicating the black-boxes behavior [9]. Moreover, transparent boolean classifiers have an added advantage of efficiency where even large boolean formulas can potentially be made extremely scalable by implementing them in hardware. Not to mention they can also be traced efficiently to determine individual classifications [23].

In summary, we have proposed a novel method based on a key insight to create boolean features from local contrastive explanations that lead to transparent models that in many cases are more locally consistent than competing approaches, while still having comparable accuracy. Even though we train on just features derived using these sparse explanations our models are comparable in accuracy to standardly trained models. In the future, we would like to build other classifiers (viz. weighted rule sets) using our boolean features. Moreover, we would also like to study the theoretically reasons behind the good generalization and local consistency provided by our method. We conjecture that this has connections to the stability results shown for stochastic gradient descent in deep learning settings, given that the local explanation method primarily relies on gradient descent. We would also like to experiment on other modalities than tabular, such as images and text, where local explanations are generated using high level features [17] making our approach directly applicable.

## Broader Impact

Explainable AI (XAI) has gained a lot of traction in industry and government given the proliferation of black-box models such as neural networks. The General Data Protection Regulation (GDPR) [30] passed in Europe requires automated systems making decisions that affect humans to be able to explain themselves. There are mainly two types of explainability: local and global. Local explainability is about per sample explanations, while global explainability is about understanding the whole model. Although significant amount of work has been done for each type, there is little work that tries to combine these two paradigms in an attempt to create global models that are also locally consistent.

Our work tries to bridge this gap for contrastive/counterfactual explanations which have been deemed as being one of the most important parts of an explanation [18]. The benefit of our method is thus that one can create globally transparent models that are locally consistent more than other schemes. This can be beneficial in appropriating trust in the black-box model with more confidence. The risks are similar to other methods that build global models (viz. distillation, profweight) that such models are still proxy models and hence, may not entirely replicate the reasoning done by the black-box model they are trying to explain.

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
