[Supplementary Material]

# Supplement

October 22, 2020

Table 1: Dataset characteristics, where $d$ is the dimensionality.

| Dataset | Num points | $d$ | # of Classes | Domain |
|---|---|---|---|---|
| Sky Survey | 10000 | 17 | 3 | Astronomy |
| Credit Card | 30000 | 24 | 2 | Finance |
| WDBC | 569 | 31 | 2 | Healthcare |
| Diabetes | 480559 | 20 | 2 | Healthcare |
| Magic | 19020 | 11 | 2 | Astronomy |
| Waveform | 5000 | 21 | 3 | Signal Proc. |

Table 2: Hyper parmeters, where $N + 1$ is the number of grid points and $\Delta$ is the skip parameter

| Dataset | $N + 1$ | $\Delta$ |
|---|---|---|
| Sky Survey | 30 | 3 |
| Credit Card | 30 | 3 |
| WDBC | 10 | 4 |
| Diabetes | 10 | 2 |
| Magic | 30 | 3 |
| Waveform | 20 | 3 |

# 1 Deep learning architectures

All the architectures were built using Keras. The loss is categorical cross entropy, adam optimizer and learning rate of 0.001 was used

## 1.1 Skyserver

```
1    model.add(Dense(128, activation='relu', input_dim=input_shape))
2    model.add(Dropout(0.5))
3    model.add(Dense(256, activation='relu'))
```

```
4    model.add(Dense(256, activation='relu'))
5    model.add(Dropout(0.5))
6    model.add(Dense(128, activation='relu'))
7    model.add(Dropout(0.5))
8    model.add(Dense(64, activation='relu'))
9    model.add(Dropout(0.5))
10   model.add(Dense(output_shape, activation='softmax'))
```

## 1.2 Credit Card

```
1    model = Sequential()
2    model.add(Dense(25,
3    kernel_initializer=keras.initializers.glorot_normal(seed=0),
     kernel_regularizer=keras.regularizers.l2(1e-4)))
4    model.add(Activation('relu'))
5    model.add(Dense(10, kernel_initializer=keras.initializers.
     glorot_normal(seed=0))), kernel_regularizer=keras.regularizers.
     l2(1e-4)))
6    model.add(BatchNormalization())
7    model.add(Activation('relu'))
8    model.add(Dropout(0.3))
9    model.add(Dense(num_classes, kernel_initializer=keras.
     initializers.glorot_normal(seed=0), activation='softmax'))
```

## 1.3 WDBC

```
1    model = Sequential()
2    model.add(Dense(20, kernel_initializer=keras.initializers.
     glorot_normal(seed=5), activation='relu'))
3    model.add(Dense(10, kernel_initializer=keras.initializers.
     glorot_normal(seed=5), activation='relu'))
4    model.add(Dense(num_classes, kernel_initializer=keras.
     initializers.glorot_normal(seed=5)))#, activation='softmax'))
5    model.add(Activation('softmax'))
```

## 1.4 Diabetes

```
1    model = Sequential()
2    model.add(Dense(35, kernel_initializer=keras.initializers.
     glorot_normal(seed=5)))
3    model.add(Activation('relu'))
4    model.add(Dense(10, kernel_initializer=keras.initializers.
     glorot_normal(seed=5)))
5    model.add(BatchNormalization())
6    model.add(Activation('relu'))
7    model.add(Dense(num_classes, kernel_initializer=keras.
     initializers.glorot_normal(seed=5)))#, activation='softmax'))
8    model.add(Dropout(0.5))
9    model.add(Activation('softmax'))
```

## 1.5 Waveform

```
1    model = Sequential()
2    model.add(Dense(15, kernel_initializer=keras.initializers.
     glorot_normal(seed=0), activation='relu'))
```

```
3    model.add(Dense(10, kernel_initializer=keras.initializers.
     glorot_normal(seed=0), activation='relu'))
4    model.add(Dropout(0.2))
5    model.add(Dense(num_classes, kernel_initializer=keras.
     initializers.glorot_normal(seed=0), activation='softmax'))
```

### 1.6 Magic

```
1    model = Sequential()
2    model.add(Dense(40, kernel_initializer=keras.initializers.
     glorot_normal(seed=0), activation='relu'))
3    model.add(Dense(25, kernel_initializer=keras.initializers.
     glorot_normal(seed=0), activation='relu'))
4    model.add(Dense(10, kernel_initializer=keras.initializers.
     glorot_normal(seed=0), activation='relu'))
5    model.add(Dropout(0.2))
6    model.add(Dense(num_classes, kernel_initializer=keras.
     initializers.glorot_normal(seed=0), activation='softmax'))
```

## 2 GBFL top 5 features

Listing 1: We present below the top 5 boolean rule based features used by Logistic regression (with L1 penalty) on the Sky Survey Dataset along with their feature importances.

GBFL rank 1 feature

1.51 >= dirf1 >= 0.0 & 0.66 >= dirf2 >= 0.0 &
0.26 >= dirf3 >= 0.0 & 629.05 >= fiberid >= 419.36 &
0.80 >= redshift >= 0.0 &  233.35 >= ra >= 137.26 &
57481 >= mjd >= 42354.42 & 14.42 >= dec >= 0.0 &
1770.52 >= plate >= 0.0

GBFL rank 2 feature

2.53 >= dirf1 >= 0.50 &
0.49 >= dirf2 >= 0.0 &
0.35 >= dirf3 >= 0.0 &
2213.15 >= plate >= 442.63 &
14.42 >= dec >= 0.0 &
0.80 redshift >= 0.0 &
262.10 >= fiberid >= 52.42 &
164.72 >= ra >= 109.81 &
57481 >= mjd >= 42354.42

GBFL rank 3 feature

1.51 >= dirf1 >= 0.0 &
0.49 >= dirf2 >= 0.0 &

```
0.35 >= dirf3 >= 0.0 &
260.81 >= ra >= 233.35 &
0.80 >= redshift >= 0.0 &
524.21 >= fiberid >= 157.26 &
14.42 >= dec >= 0.0 &
1770.52 >= plate >= 0.0 &
57481 >= mjd >= 42354.42

GBFL rank 4 feature

2.53 >= dirf1 >= 0.50 &
0.49 >= dirf2 >= 0.0 &
0.44 >= dirf3 >= 0.089 &
576.63 >= fiberid >= 366.94 &
260.81 >= ra >= 233.35 &
14.42 >= dec >= 0.0 &
0.80 >= redshift >= 0.0 &
1770.52 >= plate >= 0.0 &
57481.0 >= mjd >= 42354.42

GBFL rank 5 feature

2.53 >= dirf1 >= 0.50 &
0.49 >= dirf2 >= 0.0 &
0.35 >= dirf3 >= 0.0 &
10.82 >= dec >= 0.0 &
52.42 >= fiberid >= 0.0 &
178.44 >= ra >= 123.54 &
0.80 >= redshift >= 0.0 &
1770.52 >= plate >= 0.0 &
57481.0 >= mjd >= 42354.42
```

Listing 2: We present below the top 5 boolean rule based features used by the decision tree on the WDBC dataset ranked by their importance.

```
GBFL rank 1 feature

3575.86>=n2_area>=863.33 &
36.04>=n2_radius>=17.29 &
n1_fractald<=0.01 & n0_concavity<=0.28 &
n1_area<=363.87 & n1_compactness<=0.09 &
n1_concavepts<=0.03 & n0_symmetry>=0.13 &
n2_concavity<=0.83 & n2_fractald<=0.15 &
n0_area<=2108.08 & n0_smoothness<=0.14 &
n0_fractald>=0.04 & n0_concavepts<=0.16 &
n2_texture<=43.28 & n2_smoothness<=0.19 &
n0_perimeter<=188.5 & n2_symmetry>=0.15 &
```

n2_concavepts <=0.27 & n0_texture >=9.71 &
n0_compactness <=0.29 & n1_radius >=0.11 &
n1_texture >=0.36 & n1_perimeter >=0.75 &
n1_smoothness>=0 & n1_concavity>=0 &
n1_symmetry>=0 & n2_perimeter <=251.2 &
n0_radius >=10.5 & n2_compactness <=0.71

GBFL rank 2 feature

n0_concavity >=0.07 & n2_concavepts >=0.13 &
7.22<=n1_area <=274.71 &
0<=n1_fractald <=0.01 &
185.2<=n2_area <=2219.6 &
n0_perimeter <=140.26 &
0<=n1_compactness <=0.09 &
0<=n1_concavepts <=0.03 &
7.93<=n2_radius <=26.66 &
0.01<=n0_compactness <=0.29 &
0<=n0_concavity <=0.35 &
12.02<=n2_texture <=43.28 &
0.02<=n2_compactness <=0.88 &
0.05<=n2_fractald <=0.18 &
0.07<=n0_smoothness <=0.16 &
0.12<=n2_smoothness <=0.22 &
0.2<=n2_concavity <=1.25 &
0.09<=n2_concavepts <=0.27 &
21.06>=n0_radius >=6.98 &
n0_texture >=9.71 & n0_concavepts >=0.0 &
1322.25>=n0_area >=143.5 &
n0_fractald >=0.04 & 1.49>=n1_radius >=0.11 &
2.62>=n1_texture >=0.36 &
11.36>=n1_perimeter >=0.75 &
n1_smoothness>=0 & n1_concavity>=0 &
0.04>=n1_symmetry>=0 &
n2_perimeter >=50.41 &
n2_symmetry >=0.15 & n0_symmetry >=0.13

GBFL rank 3 feature

n0_concavity >=0.07 & n0_texture >=19.56 &
n1_area <=274.71 & n1_compactness <=0.06 &
n1_fractald <=0.01 & n2_compactness <=0.54 &
n2_fractald <=0.13 & n0_compactness <=0.23 &
n1_concavepts <=0.03 & n2_area <=2897.73 &
n2_concavity <=0.83 & n0_area <=2108.08 &
n0_smoothness <=0.14 & n0_concavity <=0.35 &
n0_concavepts <=0.16 & n1_smoothness <=0.01 &
n2_radius <=31.35 & n0_texture <=39.28 &

n0_perimeter <=188.5 & n2_texture <=49.54 &
n2_smoothness <=0.2226 & n0_symmetry >=0.106 &
n0_fractald >=0.04 & n1_radius >=0.11 &
n1_texture >=0.36 & n1_perimeter >=0.75 &
n1_concavity >=0 & n1_symmetry >=0 &
n2_perimeter >=50.41 & n2_symmetry >=0.15 &
n0_radius >=10.50 & n2_concavepts >=0.04

GBFL rank 4 feature

nucleus2_area >= 863.33 & nucleus2_radius >= 17.29 &
nucleus1_compactness <= 0.06 & nucleus1_fractal_dim <= 0.01 &
nucleus2_fractal_dim <= 0.13 & nucleus1_area <= 363.87 &
nucleus1_concave_pts <= 0.03 & nucleus2_compactness <= 0.71 &
nucleus0_smoothness <= 0.14 & nucleus0_compactness <= 0.29 &
nucleus2_texture <= 43.28 & nucleus2_concavity <= 1.04 &
nucleus0_perimeter <= 188.5 & nucleus0_area <= 2501.0 &
nucleus0_concavity <= 0.42 & nucleus0_concave_pts <= 0.20 &
nucleus2_radius <= 36.04 & nucleus2_area <= 4254.0 &
nucleus2_smoothness <= 0.22 & nucleus0_texture >= 9.71 &
nucleus0_symmetry >= 0.10 & nucleus0_fractal_dim >= 0.04 &
nucleus1_radius >= 0.11 & nucleus1_texture >= 0.36 &
nucleus1_perimeter >= 0.75 & nucleus1_smoothness >= 0 &
nucleus1_concavity >= 0 & nucleus1_symmetry >= 0 &
nucleus2_symmetry >= 0.15 & nucleus0_radius >= 14.02 &
nucleus2_perimeter >= 117.33 & nucleus2_concave_pts >= 0.09

GBFL rank 5 feature

nucleus2_concave_pts >= 0.13 & 7.22 <= nucleus1_area <= 274.71 &
0 <= nucleus1_fractal_dim <= 0.01 & 185.2 <= nucleus2_area <= 2219.6 &
0.01 <= nucleus0_compactness <= 0.23 & 0 <= nucleus0_concavity <= 0.28 &
0 <= nucleus0_concave_pts <= 0.13 & 0 <= nucleus1_smoothness <= 0.01 &
0 <= nucleus1_compactness <= 0.09 & 0 <= nucleus1_concave_pts <= 0.03 &
7.93 <= nucleus2_radius <= 26.66 & 0.02 <= nucleus2_compactness <= 0.71 &
0 <= nucleus2_concavity <= 0.83 & 0.05 <= nucleus2_fractal_dim <= 0.15 &
43.79 <= nucleus0_perimeter <= 164.38 & 0.05 <= nucleus0_smoothness <= 0.14 &
0.07 <= nucleus2_smoothness <= 0.19 & 18.27 <= nucleus2_texture <= 49.54 &
0.04 <= nucleus2_concave_pts <= 0.27 & nucleus0_radius >= 6.98 &
nucleus0_texture >= 9.71 & nucleus0_area >= 143.5 &
nucleus0_symmetry >= 0.10 & nucleus0_fractal_dim >= 0.04 &
1.49 >= nucleus1_radius >= 0.11 & nucleus1_texture >= 0.36 &
nucleus1_perimeter >= 0.75 & nucleus1_concavity >= 0.0 &
0.04 >= nucleus1_symmetry >= 0 & nucleus2_perimeter >= 50.41 &
0.42 >= nucleus2_symmetry >= 0.15

Listing 3: Top 5 boolean rule based features used by the decision tree on the
Magic dataset ranked by their importance.

GBFL rank 1 feature

```
fSize >= 2.937951724137931 &
fSize <= 3.629234482758621 &
fAlpha <= 6.206896551724138 &
fLength >= 0.0 &
fWidth >= 0.0 &
fM3Long >= 0.0 &
fAlpha >= 0.0
```

GBFL rank 2 feature

```
fM3Long <= 0.0 &
fAlpha <= 9.310344827586206 &
fLength >= 0.0 &
fWidth >= 0.0 &
fAlpha >= 0.0 &
fSize >= 2.073848275862069
```

GBFL rank 3 feature

```
fSize >= 2.7651310344827587 &
fAlpha <= 34.13793103448276 &
fSize <= 3.456413793103448 &
fWidth <= 15.207889655172414 &
fLength >= 0.0 &
fWidth >= 0.0 &
fM3Long >= 0.0
```

GBFL rank 4 feature

```
fWidth >= 30.415779310344828 &
fSize >= 2.937951724137931 &
fWidth <= 60.831558620689655 &
fSize <= 3.629234482758621 &
fM3Long <= 0.0 &
fLength >= 0.0 &
fAlpha >= 12.413793103448276
```

GBFL rank 5 features

```
fM3Long >= 15.961793103448276 &
fM3Long <= 47.88537931034483 &
fWidth <= 7.603944827586207 &
fLength <= 34.57003448275862 &
fAlpha <= 21.724137931034484 &
fLength >= 0.0 &
fWidth >= 0.0 &
fAlpha >= 9.310344827586206 &
fSize >= 2.073848275862069
```

**Remark:** Although, the method in [1] does not really perform the constrained optimization but uses regularization like 'Elasticnet' penalty to impose sparsity, we will assume that our PPs and PNs are the result of these optimizations just for simplicity of exposition. The only difference is that the sparsity $k$ cannot be pre-determined but is typically a constant for many training samples in practice.

# References

[1] A. Dhurandhar, T. Pedapati, A. Balakrishnan, P.-Y. Chen, K. Shanmugam, and R. Puri. Model agnostic contrastive explanations for structured data. *arxiv*, 2019. 2