[Reviews · NeurIPS 2020]

Review 1

Summary and Contributions: In this paper, the authors observed and addressed the problem that the predictions made by global proxy models used to explain black-box models do not agree with the explanations provided by local explanation models (e.g., in the sense that global proxy models may predict the same for the original input and the "pertinent-negative" input that results from the smallest perturbation of the original input that should change the class label). In particular, the contributions of the paper are: (1) The authors proposed a local-consistency loss function and a local-consistency metric, to measure if proxy models are consistent with the local explanations (i.e. if proxy models predict the same for the original input x and its "pertinent-positive" p(x), and if proxy models predict differently for x and its "pertinent-negative" n(x); (2) The authors proposed an algorithm to learn a set of boolean conjunctive clauses that capture the ranges of feature values for each training instance; (3) The authors proposed to learn proxy models (e.g., decision trees) on the set of boolean clauses, and the resulting models are both transparent and consistent with local explanations.

Strengths: Soundness of the claims (theoretical grounding, empirical evaluation): + The local consistency metric is a sound measure for measuring the consistency of proxy global models with respect to local perturbations. + The quantitative evaluation of the local consistency of various proxy models is convincing, and does shows that the author's method improves local consistency of proxy global models. Significance and novelty: + The idea of building global explanation (proxy) models that also behave accurately on small perturbations of input is significant. + The local consistency metric and the algorithm for generating the set of boolean clauses is novel. Relevance: + The problem of building explanation models for black-box models is relevant to explainable machine learning.

Weaknesses: Soundness of the claims (theoretical grounding, empirical evaluation): - The proposed method requires generating one boolean conjunctive clause for every training sample, and each clause requires going through every input feature. Does the method scale well to large datasets? - The authors used the set of boolean conjunctive clauses as the "features" for training the final proxy model (decision tree). This could result in: (1) long redundant "features" whose boolean conditions could overlap among each other, and (2) the final proxy decision tree whose internal nodes correspond to long conjunctive conditions (as in Listing 1). This could severely harm the interpretability of the final proxy model. Significance and novelty: - The long "features" from the set of boolean clauses, and the proxy decision tree built on such "features" are not very interpretable and could limit the use of the proposed method.

Correctness: The claims and method are generally correct. However, I would dispute the claim that the proposed method result in proxy models (decision trees) that are "reasonably easy to parse," because the features used by the proxy decision trees are long conjunctive conditions and could have many repetitions of conditions that made the models hard to parse.

Clarity: The paper is clearly written.

Relation to Prior Work: The differences of this work from prior work is clearly discussed.

Reproducibility: Yes

Additional Feedback: Suggestions for improvement: If the authors can think of a way to avoid using the long conjunctive "features" from the set of boolean clauses as the input features for the final proxy model, the work will be significantly better. ********************************* After reading the author feedback, I understand that the length of the conjunctive clauses in a feature is determined by the sparsity of the PPs and PNs generated. Are there ways to improve the sparsity of the conjunctive clauses, despite the complexity of the PPs and PNs? In other words, is it possible to drop some clauses in a conjunctive feature, while not harming the consistency and the accuracy too much? The paper will substantially benefit from (at least) some attempts to reduce the length of the long conjunctive "features" the authors are currently using.


Review 2

Summary and Contributions: This work proposed a method for training a global approximation of a black-box in such a way that it is consistent with the contrastive explanations for that black-box models. To do this it: 1) bins the features, 2) obtains the pertinent negatives and pertinent negatives for each training point, 3) converts that to a binary dataset and 4) trains an inherently interpretable model on that binary dataset. # Update based on Author Response The reviewer's primary concern was not over whether or not the proposed method's model is too complex to be interpretable (which is a possible point of concern, the argument used by [23] seems to run against the arguments of other work) but whether or not the proposed model is significantly more complicated than the baseline (eg, it certainly appears to have more parameters). This is not addressed by the author response.

Strengths: The idea of not only wanting to approximate the black box model but also be consistent with its local behavior is interesting and potentially very useful because it combines two types of analysis. The baselines and experimental setup are setup well.

Weaknesses: The primary concern with the results is whether or not the decision tree learned by GBFL is significantly more complicated than the ones learned by the standard, distillation, or augmentation approaches. Although the are all of depth at most five, the features used by GBFL each involve all of the dataset features with an upper and a lower bound. If this concern can be addressed, the reviewer thinks this would be a much stronger submission.

Correctness: The claims and methodology are seem to be largely correct. The description of Equation 1 ("Being sparse, it is expected to have lesser variation away from the base values than x.") might be inaccurate if delta < -|x| and x >= b.

Clarity: Generally, the paper is easy to read with the exception being the description of how the new binary features are constructed. Adding a simple 1-d toy example would help with this. Relatedly, Figure 1 is very busy and it is a bit difficult to separate all of the pieces.

Relation to Prior Work: The connections to prior work are clear

Reproducibility: Yes

Additional Feedback: How larges does |F| get? It sees like it could be a significant portion of the size of the dataset. In addition to the C_TB^{PP/PN} metrics, it would be nice to have the portion of the time that the interpretable model matches the black-box model on the original data (ie, normal global approximation accuracy).


Review 3

Summary and Contributions: This paper aims to make the transparent model consistent with the explanation of black-box models. Authors propose to use the local contrastive explanation of DNNs to create custom boolean features, which are used to train a transparent model. This paper also introduces a natural local consistency metric that quantifies if the local explanations and predictions of the black-box model are also consistent with the proxy global transparent model.

Strengths: This paper is well-motivated, and it is interesting to produce a transparent global model that is simultaneously accurate and consistent with the local contrastive explanations of the black-box model.

Weaknesses: 1. In Eq. (1) and Eq. (2), how are the upper bounds and lower bounds determined? Do you set the bound manually first and then calculate $p(x)$ and $n(x)$, or obtain bounds through the calculation of $p(x)$ and $n(x)$? If you obtain bounds through the calculation, then the definition in Eq. (1) and Eq. (2) is problematic. If the bounds are set manually, then what is the form of bounds? how do you determine the bound with high dimensions, and how can you guarantee that there exists pertinent negative vector within the bounds? 2. There is no rigorous formulation and justification for the proposed method. Besides, why not use the knowledge distillation to learn the transparent models? It seems that the knowledge distillation is more direct. 3. In Eq. (3), why do you require that the output of the transparent model for the PN is different from the output of the black-box for the original input? If you want to ensure the consistency between two models, for the PN, the output of two models should be the same. 4. In Eq. (5), only one pertinent positive data point is used. However, there are many data points that satisfy the Eq. (1) and Eq. (2). These data points contribute equally to the Eq. (3), but there perturbation directions are different. Eq. (5) doesn’t include the exhaustive test over the diversity of all these data points. 5. Authors are making a circular arguments when verifying the effectiveness of the proposed method. They use the pertinent positive and pertinent negative to define the metric for the consistency, as well as use them to train the transparent global model. They are all based on the same ideas, so the comparison in Table 1 is questionable. 6. In Experiments, the datasets used are in low dimension. Authors should conduct experiments on more general datasets, e.g. MNIST, to show their effectiveness. 7. In section 4, PP and PNs in Eq. (5) are defined as vectors (features) with high dimensions. However, in Listing 1, the condition bounds are scalars. What is the correct form of PP and PNs? 8. The results shown in the Listing 1 is still unclear enough. Authors should investigate further to obtain an explanation in the semantic level. 9. The paper is not well-organized and hard to understand. The symbol used in Figure 1 is not introduced specific, it is hard to understand the Figure 1 even after I read the paper.

Correctness: Questionable. Please see above comments.

Clarity: No. Please see above comments.

Relation to Prior Work: Yes

Reproducibility: No

Additional Feedback:


Review 4

Summary and Contributions: The paper proposes to train a locally consistent explanation model for a given model as a decision tree over boolean features. The goal is that the explanation model is locally truthful to the blackbox model. They bring the idea that one can add to every point x two more points, one adversarial-type, the PN, and one of minimal support type, the PP, and to enforce correct prediction of the explanation model also on the PN and PP.

Strengths: The idea is neat to construct a PN and PP along a sample and to construct boolean clauses from it for training constraints. Adding the augmentation baseline as data samples for comparison is appreciated.

Weaknesses: One weakness is the fixation on a single PP and PN per sample. In principle there might be several solutions with very similar objective function values for PN(x) or PP (x), which are however changed in different subsets of dimensions of x. These different solutions can be hard to compare/rank against each other if feature dimensions have incompatible physical dimensions e.g. if i reduce age by 5 years or income by 3000$ to get to a PN - what is closer to x then ? the authors replied in their rebuttal that this is given, well, it is half an answer, and leaves an open question / free parameter. The work makes the assumption that x and the base vector share the same class, when defining PP and PN. This makes a kind of limiting assumption on the structure of the model to be explained, which may sometimes be not applicable (e.g. a class like: outlier where the outlier space is unbounded), sometimes can be hard to find (e.g. classes with are defined by a union of disjoint gaussians, then one needs one b for each gaussian), and this assumption is likely not needed, when PP and PN are defined relative to x instead of relative to a unique b. the authors explained that this is not the case. point taken. This is not a method that seems to have a good runtime for high-dimensional problems say images or videos. this is not affecting the score of the reviewer though, as the method is applicable for problems in finance and tabular healthcare data. the scalability point remains after the rebuttal, as thee authors suggest simple models to be trained.

Correctness: eq. (1) seems to have a small semantical issue (also a syntactical issue with one } too much). Given that the goal for a PP is to preserve the class label of the original sample, for a large number of classes (e.g. 1000), the value of the highest wrong class max_{\tilde{y} \new y } C(\delta, \tilde{y}) is always low, but the value of the correct class C(\delta,y) can still drop a lot, if the probability is redistributed equally among all wrong classes. Thus is seems to be more sensible to look at C(x,y)-C(\delta,y), and keep that small together with keeping the zero-norm-k small.

Clarity: it is acceptably writtten.

Relation to Prior Work: reviewer is not from the field of reasoning with boolean clauses. Thus no strong opinion.

Reproducibility: Yes

Additional Feedback: I think it is worth to evaluate how the performance changes when multiple PN or PP are added and on problems where feature dimensions are not comparable against each other. scalability/runtime with dimensionality would be interesting. likely it is not that fast. evaluation on outlier points. a partial ablation study would be interesting: e.g. another binning, or not-grid-clamped clauses for PP and PN. Have the authors adequately addressed the broader impact of their work, including potential negative ethical and societal implications of their work? -- did not enter the score, likely this work has low ethical risk.

[Author Response · NeurIPS 2020]

We thank the reviewers for their effort. We are pleased that everyone appreciated the novelty of the idea and significance
of the problem setting of building locally consistent transparent models. We now address your specific concerns.

**R1** and **R2** *Long rules and interpretability*: Firstly, it is not true that the clauses would involve all the features, as
mentioned in the paper, our Boolean clauses are 2k sparse given k sparse local explanations i.e. PPs and PNs. So the
sparsity of our clauses depends on the sparsity of the explanations which a contrastive method provides, where our
contribution does not add any additional complexity. For example, this is seen on the Sky Survey dataset in the paper
where the dataset has 17 features but the rules contain only 9 features. Secondly, the issue of many conjunctions and
redundant features etc could arise even with known greedy algorithms for learning such structures. In fact, Rudin
(citation [23] in the paper) argues that even if conjunctive rules are pages long, a person looking for an explanation can
relatively easily "locate by inspection" a sufficiently sparse explanation by navigating the formula. This is cited as the
reason for general desirability of rule/tree based classifiers that may not be succinct.

**R1, R2, R3** and **R4** *Scalability*: If the dataset is high dimensional, as mentioned above, our method being simply a
function of the PPs and PNs which are typically sparse we should be able to scale. If the number of points is large
one can do random sampling or using prototype selection methods to control size of $F$. More importantly though,
this size is not a bottleneck since we suggest training simple models on it such as a small decision tree or L1-logistic
regression which scale well even with many features and would end up choosing only a few of these rules. The most
time consuming part is really obtaining PNs from a contrastive method as it is a non-convex optimization problem.
Although, this too could be parrallelized across different (batches of) data points.

**R2** *Correctness of description of Eq 1*: The description is correct since, $\delta_j$ always lies between $x_j$ and $b_j$ no matter the
sign. So $\delta_j$ can be $< -|x_j|$ only if $|b_j| > |x_j|$ and, $b_j$ and $x_j$ have opposite signs i.e. one is positive and the other is
negative. Here again $\delta_j$ would be closer to $b_j$ than $x_j$ is to it.

**R2** *Clarity of binning*: Please see Figure 1 in supplement for a toy example depicting the whole process. We will move
this into the main paper in the final version, since NeurIPS typically allows an extra page.

**R2** *Reporting fidelity*: Yes, we will add a row in Table 1 showcasing fidelity of the transparent model to the black-box.
Although, the last 5 rows in Table 1 which show Test accuracy hint towards what the fidelities might be.

**R3** *Regarding the form of the bounds and how they are set*: The bounds $L_j$ and $U_j$ in Eq.(1) and Eq.(2) are set manually
and its basically a bound on the domain we know. For example, features like "Age" cannot be negative and will have a
lower bound of 0, while for gray scale images the values would be 0 and 1 for all pixels. There is no guarantee that one
can find a PN vector for every data point using these contrastive methods. However, that is independent of our current
contribution and our method can create rules even if we have no PNs. This can be confirmed by looking at the Magic
and Diabetes datasets in Table 1 where we can still create rules eventhough no PNs were found for those datasets.

**R3** *Why not Knowledge distillation*: Please look at Table 1 where we perform favorably to knowledge distillation.

**R3** *Regarding Eq 3*: By the definition of PN, we are aiming to generate a point $n(x)$ that is of a different class from the
original input $x$ and thus equation is correct.

**R3** *Only one PP used in Eq. 5*: Contrastive methods (citations [8] and [31] in the paper) output only a single PP and at
most one PN explanation for each data point. We use this PP and PN in Eq 5 for discretization.

**R3** *Regarding circular argument*: The local consistency metric just ensures that the predictions as well as the local
explanations (PPs and PNs) for the black-box are consistent with the transparent model. This is independent of how one
might obtain a locally consistent model. When trying to build a such a model one is free to use all the information that is
available to them. So we do not believe there is any circular argument. In fact, the Augmentation baseline we compare
with in Table 1 is more closely related to just directly using PPs and PNs, and we outperform it by a significant margin.

**R3** *Dimension of PPs and PNs in Eq (5)*: $p(x_j)$ means the j-th coordinate of the PP vector corresponding to the point $x$.
So there is no discrepancy between this and Listing 1. We will clarify.

**R4** *Selecting between PNs*: Data is typically normalized where PPs and PNs are generated by another method and
given as input to ours. So though you have a valid point that is not the focus of this submission, rather we assume
whichever local contrastive method we use has already done that for us and provided a valid local explanation. Realistic
explanations in previous works were generated using autoencoders etc. (see citation [8] in the paper).

**R4** *Assumption that $x$ and the base vector share the same class*: This is not true. There is only one base vector that is
set for the entire dataset. This may be a zero vector or one where the base value for each feature is set based on domain
knowledge (see citation [9] in the paper). Given this, the limitations you mentioned do not apply.

**R4** *More sensible to look at $C(x, y)$-$C(\delta, y)$*: Interesting suggestion! Although, the issue you mention can be addressed
(indirectly) in Eq. 1 by having a large enough $\kappa$, which controls the margin between $y$ and the next most likely class.

[Meta-Review · NeurIPS 2020]

Through an active discussion, it was clear that the reviewers found the proposed ideas to be quite valuable, and the evaluation to be quite convincing. However, the scope of the work needs to be considerably clarified in the subsequent versions, for example, make it obvious that it will only work on tabular datasets, and without long conjunctions of features in the model. With appropriate framing (which we strongly encourage the authors to do for the final version), this paper will likely have much more impact.